# Transcriptome Analysis Revealed Potential Genes of Skeletal Muscle Thermogenesis in Mashen Pigs and Large White Pigs under Cold Stress

**DOI:** 10.3390/ijms242115534

**Published:** 2023-10-24

**Authors:** Wenxia Li, Yufen Chen, Yunting Zhang, Ning Zhao, Wanfeng Zhang, Mingyue Shi, Yan Zhao, Chunbo Cai, Chang Lu, Pengfei Gao, Xiaohong Guo, Bugao Li, Sung-Woo Kim, Yang Yang, Guoqing Cao

**Affiliations:** 1College of Animal Science, Shanxi Agricultural University, Jinzhong 030801, China; lwx8lois@163.com (W.L.);; 2Department of Animal Science, North Carolina State University, Raleigh, NC 27695, USA

**Keywords:** cold stress, skeletal muscle, transcriptome, Large White pig, Mashen pig, PRSS8

## Abstract

Pigs are susceptible to cold stress due to the absence of brown fat caused by the partial deletion of *uncoupling protein 1* during their evolution. Some local pig breeds in China exhibit potential cold adaptability, but research has primarily focused on fat and intestinal tissues. Skeletal muscle plays a key role in adaptive thermogenesis in mammals, yet the molecular mechanism of cold adaptation in porcine skeletal muscle remains poorly understood. This study investigated the cold adaptability of two pig breeds, Mashen pigs (MS) and Large White pigs (LW), in a four-day cold (4 °C) or normal temperature (25 °C) environment. We recorded phenotypic changes and collected blood and longissimus dorsi muscle for transcriptome sequencing. Finally, the *PRSS8* gene was randomly selected for functional exploration in porcine skeletal muscle satellite cells. A decrease in body temperature and body weight in both LW and MS pigs under cold stress, accompanied by increased shivering frequency and respiratory frequency, were observed. However, the MS pigs demonstrated stable physiological homeostasis, indicating a certain level of cold adaptability. The LW pigs primarily responded to cold stress by regulating their heat production and glycolipid energy metabolism. The MS pigs exhibited a distinct response to cold stress, involving the regulation of heat production, energy metabolism pathways, and robust mitochondrial activity, as well as a stronger immune response. Furthermore, the functional exploration of *PRSS8* in porcine skeletal muscle satellite cells revealed that it affected cellular energy metabolism and thermogenesis by regulating ERK phosphorylation. These findings shed light on the diverse transcriptional responses of skeletal muscle in LW and MS pigs under cold stress, offering valuable insights into the molecular mechanisms underlying cold adaptation in pigs.

## 1. Introduction

Cold stress is a significant environmental factor in alpine regions, imposing various adverse effects on the physiological systems of animals, including the nervous, endocrine, immune, and antioxidant systems. This situation leads to substantial threats to animal husbandry production and development [1,2]. Pigs, in particular, are highly susceptible to ambient temperature variations, with cold stress contributing to an increase in the incidence of pig diseases [3] and mortality rates [4], as well as diminishing meat quality [5]. These challenges ultimately lead to severe economic losses. The absence of *uncoupling protein 1* (*UCP1*) in pigs throughout their evolutionary history has resulted in a lack of brown adipose tissue (BAT), rendering them more cold-sensitive. However, certain pig breeds in China, such as Tibetan pigs [6] and Min pigs [7], have demonstrated innate cold adaptability due to their geographical exposure. Genetic variations among pig breeds influence their cold tolerance characteristics [8]; nonetheless, the precise molecular mechanisms of cold tolerance in pigs remain elusive. Notably, when Tibetan pigs were acutely exposed to 4 °C for 4 h, they showed heightened expression of *uncoupling protein 3* (*UCP3*) and a beige-like appearance of their white fat, suggesting the crucial role of fat metabolism in cold adaptation [9]. Furthermore, Yorkshire pigs under chronic cold stress exhibited visible pathological changes in their jejunal and ileal mucosa, with enhanced expression of *toll*-*like receptor 4* (*TLR4*), *myeloid differentiation primary response gene 88* (*MyD88*), and *nod*-*like receptor pyrin domain containing 3* (*NLRP3*). In contrast, Min pigs did not exhibit such harmful consequences under similar conditions [10].

The Mashen (MS) pig, a local breed found in northern China, is renowned for its strong resistance to stress, high-quality meat, high fecundity, slow growth rate, and ability to thrive on roughage [11,12]. Recent studies have also found that MS pigs exhibit a certain level of cold adaptability. When mice were transplanted with MS pig microbiota and subjected to 21 days of cold exposure at 4 °C, they showed a more stable body temperature and a better-preserved intestinal structure compared to mice transplanted with Duroc-Landrace-Yorkshire (DLY) pig microbiota. Additionally, the expression of *UCP1*, *carnitine palmitoyltransferase 1B* (*CPT1B*), and *peroxisome proliferator*-*activated receptor*-*gamma coactivator 1 alpha (PGC*-*1α*) significantly increased in the MS pig microbiota-transplanted mice (*p* < 0.05) [13]. Moreover, in low-temperature environments, local MS pigs demonstrated a stronger ability to maintain their intestinal physiological function compared to Large White (LW) and Jinfen White pigs. Their duodenal esophageal wall also exhibited higher trypsin activity, promoting an inflammatory response [14]. The current research on cold adaptation in pigs chiefly centers on their fat and intestines, and further exploration is required to understand the cold adaptation mechanism in other organs.

Skeletal muscle accounts for approximately 40% of the total body weight in animals [15] and serves as the largest metabolic organ [16]. In response to cold stress, shivering thermogenesis (ST) in skeletal muscle is the primary means of thermogenesis, involving spontaneous repeated muscle contractions and relaxations [17,18]. This process relies mainly on actin, myosin, and adenosine triphosphate (ATP) hydrolysis [19]. Additionally, non-shivering thermogenesis (NST) in skeletal muscle is a major contributor to cold adaptation and involves the ineffective Ca^2+^ cycling of sarcoplasmic reticulum Ca^2+^-ATPase (SERCA) [20] and the uncoupling of the mitochondrial inner membrane [21]. Studies on *Lasiopodomys brandtii* have shown that the removal of interscapular BAT resulted in enhanced thermogenesis in skeletal muscle under a 4 °C environment for 28 days. This enhancement was characterized by an amplified mitochondrial quantity and metabolic activity, along with elevated levels of sarcolipin (SLN) and sarcoplasmic reticulum Ca^2+^-dependent ATP [22]. Furthermore, newborn wild boars exposed to cold environments during their first five days after birth exhibited an increased contribution of NST, accompanied by age-related increases in body temperature, SERCA activity, and expression of *SERCA1A* and *SLN* in skeletal muscle tissue [23]. In a study involving zinc-alpha-2-glycoprotein knockout (ZAG-KO) mice and wild-type (WT) mice exposed to a low temperature (6 °C) for one week, it was found that cold stress increased the expression levels of lipolysis-related proteins (ATGL and p-HSL) and energy metabolism-related proteins (PGC-1α, UCP2, UCP3, and COX1) in the gastrocnemius muscle of WT mice [24]. Meanwhile, another study discovered that *UCP3* overexpression in mouse skeletal muscle resulted in the reduced efficiency of muscle mitochondria, leading to a 42% reduction in the ratio of ATP synthesis to mitochondrial oxidation [25]. This indicated that *UCP3* was subject to complex regulatory mechanisms in NST. To support an increase in animal heat production, the energy metabolism, especially glucose and lipid metabolism, needs improvement [26,27]. ST primarily involves glycogen decomposition in skeletal muscle, with the rate of glycogen utilization correlating with the muscle contraction strength [28,29]. The screening of differentially expressed genes (DEGs) from rat iliopsoas muscles exposed to ultra-low temperatures, indicated their involvement in muscle regeneration, tissue repair, and lipid metabolism [30]. Based on these findings, we infer that pigs respond to cold stress by upregulating their glycolipid energy metabolism, which provides sufficient energy for skeletal muscle thermogenesis.

In this study, a local breed, MS pigs, and an imported breed, LW pigs, were used as experimental subjects, exposed to either low temperature (4 °C) or normal temperature (25 °C) conditions for a period of four days. Blood biochemical parameters were measured using enzyme-linked immunosorbent assay (ELISA), and an ultrastructure and an enzyme activity assessment of longissimus dorsi muscle were performed using transmission electron microscopy and enzyme histochemical staining, respectively. The DEGs and enriched pathways in response to cold stress were screened and analyzed to identify the ones shared by both LW and MS pigs, as well as those unique to MS pigs based on transcriptome sequencing in the longissimus dorsi muscle of both pig breeds. This study will provide a comprehensive analysis of the distinct physiological symptoms and transcriptional responses between LW and MS pigs under cold stress, with a specific focus on the screening of cold-response candidate genes in pigs and preliminary verification to reveal their functions. The outcomes will provide valuable insights into the cold adaptation mechanisms of different pig breeds and contribute to future research on breeding cold-resistant pigs.

## 2. Results

### 2.1. Different Physiological Response Patterns of Large White Pigs and Mashen Pigs under Cold Stress

LW and MS pigs were subjected to a normal temperature (25 °C) and low temperature (4 °C) for a duration of four days. During the experiment, the body temperature of both pig breeds at 4 °C was significantly lower (*p* < 0.05) than at 25 °C (Figure 1A). However, the body temperature deviation in the MS pigs was most significantly (*p* < 0.01) lower than that of the LW pigs in the first two days of cold stress (Figure 1B). Additionally, the body temperature deviation of MS pigs was significantly (*p* < 0.05) lower than that of LW pigs on the fourth day of cold stress. No significant difference in respiratory rate was observed between the two breeds at either 25 °C or 4 °C (Figure 1C). During the cold treatment, the weight of the MS pigs decreased by 2.98 kg, representing a 7.2% reduction compared to their initial weight (Appendix A). The weight of the LW pigs decreased by 1.29 kg, representing a 3.6% reduction compared to their initial weight. The statistical results of the average daily feed intake showed that temperature had a significant effect on the average daily feed intake of the two breeds of pigs (*p* < 0.05). The interaction between the temperature and the breed of the pigs had no significant effect on the average daily feed intake of the two breeds (Appendix A). At 4 °C, the shivering frequency of the MS pigs was significantly higher (*p* < 0.05) than that of the LW pigs on the first day and significantly lower (*p* < 0.05) than that of the LW pigs on the fourth day (Figure 1D). Interestingly, during the cold treatment, the shivering frequency of the MS pigs recovered more gently compared to the LW pigs, indicating that the MS pigs maintained more stable physiological homeostasis and exhibited better adaptability to low temperatures.

### 2.2. Serum Biochemistry and Hormone Analyses of Large White Pigs and Mashen Pigs under Cold Stress

At a temperature of 4 °C, the activities of three thermogenic enzymes, four glucose metabolism enzymes, and two mitochondrial energy metabolism enzymes were measured. It was found that the EPI and NE levels increased the most significantly (*p* < 0.01) in LW pigs under cold stress, and the Ca^2+^/Mg^2+^-ATPase levels increased significantly (*p* < 0.05) in LW pigs under cold stress. In addition, in the LW pigs, there was a notable increase (*p* < 0.01) in the glucose metabolism enzymes LDH, PFK-1, HK2, and M2-PK, as well as the mitochondrial energy metabolism enzymes SDH (*p* < 0.05) and CS (*p* < 0.01) (Figure 2A–I). However, the thermogenic and glucose metabolism enzymes in the MS pigs did not show significant changes under cold stress, except for SDH and CS, which exhibited a marked increase (*p* < 0.01). At 4 °C, the SDH and CS levels in the MS pigs were significantly higher (*p* < 0.01) than those in the LW pigs. These findings suggest that LW pigs respond to cold stress by increasing their heat production, glucose metabolism, and mitochondrial energy metabolism, whereas MS pigs primarily respond to cold stress by enhancing their mitochondrial energy metabolism, with higher levels of mitochondrial activity compared to LW pigs.

### 2.3. Effects of Cold Stress on Ultrastructure and Enzyme Activity of Longissimus Dorsi Muscle in Large White Pigs and Mashen Pigs under Cold Stress

The electron microscope analysis of pig longissimus dorsi revealed changes in the abundance and ultrastructure of mitochondria in both the LW and MS pigs under cold stress. At 4 °C, the mitochondrial abundance between skeletal muscle fibers was higher in both the LW and MS pigs compared to 25 °C (Figure 3A). However, the cristae structure of mitochondria in the MS pigs appeared more intact, while the cristae structure in the LW pigs showed damage under cold stress. Further, the MS pigs exhibited a higher number of mitochondria between their muscle fibers than the LW pigs at both 25 °C and 4 °C. SDH activity staining (Figure 3B) confirmed a significant increase (*p* < 0.01) in mitochondrial energy metabolism in both LW and MS pigs under cold stress, with the MS pigs showing the highest increase (*p* < 0.01) compared to the LW pigs. LDH activity staining (Figure 3C) revealed that the LW pigs had a significant increase (*p* < 0.01) in glycolysis metabolism under cold stress, while the MS pigs did not exhibit a significant change in glycolysis metabolism under cold stress.

### 2.4. Differential Expression Genes Analysis of Longissimus Dorsi Muscle in Large White Pigs and Mashen Pigs under Cold Stress

RNA-seq analysis was conducted on longissimus dorsi muscle samples from 12 pigs. The results indicated that the Q30 values of all filtered samples were above 90% (Appendix A). The distribution of the sequencing data was even among the 12 subjects, with no significant difference observed between different individuals (Figure 4A). The PCA plot (Figure 4B) and heat maps (Appendix A) indicated that the three biological replicates of each group were of high quality, and the samples clustered well within the four groups: normal-temperature group (LW25–vs.–MS25), low-temperature group (LW4–vs.–MS4), LW pig group (LW25–vs.–LW4), and MS pig group (MS25–vs.–MS4). In the LW pig group, a total of 750 DEGs were identified, with 374 up-regulated genes and 376 down-regulated genes (Figure 4C). Conversely, in the MS pig group, only 46 DEGs were identified, comprising 31 up-regulated genes and 15 down-regulated genes. Notably, the proportion of up-regulated and down-regulated DEGs in the LW pig group was roughly balanced, while the number of up-regulated DEGs in the MS pig group was twice that of down-regulated DEGs (Figure 4D; Appendix A). Within the overlapping region of the LW pig group and the MS pig group, 23 DEGs played a crucial role in the common cold response shared by both LW and MS pigs. On the left side of the non-coincidence region, 727 DEGs exerted regulatory roles in the specific cold response of the LW pigs, while on the right side, 23 DEGs governed the specific cold response of the MS pigs (Figure 4E; Appendix A).

The fold change in DEGs in the LW pig group and MS pig group mostly ranged from one to two times, followed by two to three times, and the least common was more than three times (Appendix A). Among the 23 common DEGs in response to cold stress shared between the LW pig group and MS pig group (Figure 4F), 19 DEGs were significantly up-regulated, and 4 DEGs were significantly down-regulated. Notably, *NR4A2* and *LEP* were the DEGs with the largest up-regulation and down-regulation folds, respectively, in both the LW pig group and MS pig group (Appendix A). *NR4A2* was up-regulated by 2.84 times in the LW pig group and 2.61 times in the MS pig group, respectively, while *LEP* was down-regulated by 3.83 times in the LW pig group and 3.55 times in the MS pig group, respectively. Additionally, there were 23 unique cold-responsive DEGs in the MS pig group (Figure 4G). Among these, *LYZ* and *RDH16* exhibited the largest up-regulation and down-regulation folds, respectively, in the MS pig group. *LYZ* was up-regulated by 3.40 times, while *RDH16* was down-regulated by 3.60 times in this group (Appendix A). Furthermore, the LW pig group had 727 unique cold-responsive DEGs, with *CR1* and *HGFAC* showing the largest up-regulation and down-regulation folds, respectively, in the LW pig group. *CR1* was up-regulated by 6.13 times, while *HGFAC* was down-regulated by 9.74 times in this group (Appendix A). 

### 2.5. Functional Annotation and Pathway Enrichment Analysis of Common and Specific Cold-Responsive Differentially Expressed Genes in Large White Pigs and Mashen Pigs under Cold Stress

The DEGs shared by both the LW and MS pigs in response to cold stress were enriched in GO terms related to heat production and energy metabolism (Figure 4H). Specifically, *ARRDC3*, *ADRB2*, *CEBPB*, and *NR1D1* were enriched in cold-induced thermogenesis (Appendix A) and were also associated with various pathways related to glycolipid energy metabolism. *SIK1* was found to be enriched in the negative regulation of metabolic processes, lipid metabolic processes, and carbohydrate biosynthetic processes. Additionally, *NR4A2*, *CIART*, *KLF10*, and *TIPARP* were enriched in the negative regulation of metabolic process, with *NR4A2* being the most up-regulated gene in both the LW and MS pigs under cold stress. Furthermore, *GPCPD1* and *SCD* were enriched in lipid metabolic processes. The KEGG pathways shared by the LW and MS pigs in response to cold stress were significantly enriched in environmental information processing, metabolism, human diseases, organismal systems, and cellular processes. These pathways included the adenosine monophosphate-activated protein kinase (AMPK) signaling pathway, non-alcoholic fatty liver disease, the JAK-STAT signaling pathway, the biosynthesis of unsaturated fatty acids, the circadian rhythm, and apoptosis-multiple species (Figure 4I). For instance, *SCD* was enriched in the AMPK signaling pathway and the biosynthesis of unsaturated fatty acids, and it was also associated with the lipid metabolism pathway in GO. *NR1D1* was enriched in the circadian rhythm, which was related to environmental adaptation, and it was also involved in the heat production and glycolipid energy metabolism pathways in GO. Moreover, *LEP* was enriched in multiple energy metabolism pathways, such as the AMPK signaling pathway, non-alcoholic fatty liver disease, and the JAK-STAT signaling pathway. Notably, *LEP* was the most down-regulated gene in both the LW and MS pigs under cold stress, and it was also enriched in the lipid metabolism pathway in GO. Based on the results of the GO and KEGG analyses, the candidate genes of LW and MS pigs in response to cold stress were identified as *ARRDC3*, *ADRB2*, *CEBPB*, *SIK1*, *NR4A2*, *GPCPD1*, *CIART*, *KLF10*, *TIPARP*, *SCD*, *NR1D1*, and *LEP*. As such, these candidate genes appear to play vital roles in regulating heat production and energy metabolism pathways during cold stress adaptation in LW and MS pigs.

The DEGs that were specific to the response of MS pigs to cold stress were significantly enriched in various pathways related to cold adaptation, including hemoglobin metabolism, adaptive immunity, fiber contraction, oxygen transport, and transport regulation (Figure 4J). Among these pathways, specific DEGs were identified, and *ALAS2* was enriched in the hemoglobin metabolic process (Appendix A). *DUSP10* and *IL18* were enriched in the adaptive immune response, *ACTC1* and *ANKRD1* were enriched in the contractile fiber part, and *ANKRD1* was enriched in the regulation of transport. Further, *HBB* was enriched in oxygen transport, while *PRSS8*, *NEFH*, and *KCNC4* were enriched in the regulation of transport. Additionally, the specific KEGG pathway for MS pigs in response to cold stress was significantly enriched in human diseases and metabolism, including pathways related to immunity and metabolism such as african trypanosomiasis, malaria, legionellosis, influenza A, and porphyrin metabolism (Figure 4K). *HBB* and *IL18* were enriched in African trypanosomiasis and malaria, respectively, and they were also involved in oxygen transport and the adaptive immune response, as identified in the GO analysis. *HSP70.2* was enriched in legionellosis and influenza A, and it has been studied extensively for its role in cold hardiness as a member of the heat shock protein family. *ALAS2* was enriched in porphyrin metabolism and was also identified in the GO analysis for its involvement in the hemoglobin metabolic process. These findings suggest that the specific response of MS pigs to cold stress is mainly achieved by participating in immune and metabolic pathways. Based on the GO and KEGG results, candidate genes that are specific to the cold stress response of the MS pigs were identified as *ACTC1*, *PRSS8*, *DUSP10*, *ANKRD1*, *NEFH*, *KCNC4*, *HBB*, *IL18*, *HSP70.2*, and *ALAS2*.

The DEGs that were specific to the response of LW pigs to cold stress were significantly enriched in pathways related to cold environment adaptation, including thermogenesis, glycolysis, and fatty acid oxidation (Appendix A). Notably, *ACSL1*, *CD36*, *ELOVL6*, and *PPARGC1A* were enriched in the cold-induced thermogenesis pathway (Appendix A). Additionally, *CPT2*, *IRS2*, *ACADM*, *HADHB*, *PCK2*, and *APOC3* were involved in the cellular lipid catabolic process, with *IRS2* and *ACADM* also being associated with glucose metabolism. Furthermore, *IGF1* and *HK2* were linked to the regulation of glucose import. The specific KEGG pathways of the LW pigs in response to cold stress included the forkhead box O (FoxO) signaling pathway, the peroxisome proliferator-activated receptor (PPAR) signaling pathway, the fatty acid metabolism, the AMPK signaling pathway, and insulin resistance, which were closely related to cold stress (Appendix A). Among these pathways, *PPARGC1A*, *CD36*, *IRS2*, *PCK2*, and *IGF1* were enriched in the AMPK signaling pathway, and *PCK2* was additionally enriched in the adipocytokine signaling pathway and insulin resistance. *ACSL1*, *CPT2*, *ACADM*, and *APOC3* were enriched in the PPAR signaling pathway, while *ELOVL6* and *HADHB* were related to fatty acid metabolism. Moreover, *IRS2* and *HK2* were associated with type II diabetes mellitus. These findings indicated that the LW pigs specifically responded to cold stress by actively participating in heat production and glycolipid metabolism pathways. Based on the GO and KEGG analysis results, the candidate genes specifically responding to cold stress in the LW pigs were inferred to be *PPARGC1A*, *CD36*, *IRS2*, *PCK2*, *IGF1*, *ACSL1*, *CPT2*, *ACADM*, *APOC3*, *IRS2*, and *HK2*.

### 2.6. Weighted Co-Expression Network Analysis of Longissimus Dorsi Muscle in Large White Pigs and Mashen Pigs under Cold Stress

Using the WGCNA method, we analyzed a total of 12,365 genes from the transcriptome data after quality control. The optimal soft threshold power of 17 was determined with a criterion of R^2^ > 0.8 to construct a scale-free network (Appendix A). Based on the gene expression patterns, we divided the modules into clusters and merged those with a similarity greater than 0.75. Ultimately, we identified a total of 16 modules, and genes with no clear co-expression trend were placed in the grey module (Appendix A and Figure 5A,B). Among these modules, the blue2 module contained the highest number of genes, with 3766 genes, while the grey module had the fewest, with 60 genes.

We calculated the correlation coefficients and *p* values between the modules and various physiological traits, including body weight, body temperature, respiratory rate, shivering frequency, EPI, NE, Ca^2+^/Mg^2+^-ATPase, LDH, PFK-1, HK2, SDH, CS, and M2-PK, using the eigenvector values of each module (Figure 5C). The modules showing a strong correlation with the target traits (R > 0.8, *p* < 0.01) were selected as the target modules representing the common response to cold stress among both pig breeds. Our analysis identified two such modules, namely the pink and yellow modules (Appendix A), which each showed a high correlation with multiple target traits. Further examination of the gene expression pattern in the heat map module revealed that the pink module exhibited a more similar expression trend between the LW and MS pigs under cold stress, suggesting a better representation of the common response to cold stress among both breeds (Figure 5D). Additionally, we observed that the blue2 module had the highest correlation with the MS-4 group, indicating its relevance to the unique response mode of the MS-4 group to cold stress. Consequently, we chose the pink and blue2 modules as the target modules for subsequent functional annotation and enrichment analysis.

### 2.7. Functional Annotation and Pathway Enrichment Analysis of the Target Modules

GO analysis of the pink module, which reflected the common response to cold stress among the LW and MS pigs, revealed significant enrichment in various biological processes, including the catabolic process, the cellular metabolic process, the generation of precursor metabolites and energy, the oxidation-reduction process, the energy derivation by oxidation of organic compounds, and macroautophagy (Figure 6A,B). Within this module, *ARRDC3*, *ADRB2*, *CEBPB*, *NR1D1*, *UCP3*, and *CD36* were enriched in adaptive thermogenesis, while *CASQ1* and *ACADM* were enriched in response to the temperature stimulus (Appendix A). Additionally, *SIK1* and *PDK2* were enriched in the regulation of gluconeogenesis, and *RCAN1*, *PPP3CC*, and *ATP1B1* were enriched in calcium-mediated signaling. In terms of the KEGG analysis, the pink module was mainly enriched in pathways such as the citrate cycle (TCA), Epstein-Barr virus infection, oxidative phosphorylation, carbon metabolism, non-alcoholic fatty liver disease, the AMPK signaling pathway, thermogenesis, and pyruvate metabolism (Figure 6C). Specifically, *CD36* and *ACADM* were enriched in the PPAR signaling pathway (Appendix A), while *RCAN1* and *PPP3CC* were enriched in Kaposi sarcoma-associated herpesvirus infection, and *ADRB2* and *ATP1B1* were enriched in the cGMP-PKG signaling pathway. Based on the combined results of the GO and KEGG analyses from the WGCNA, the candidate target genes of the pink module were identified as *ARRDC3*, *ADRB2*, *CEBPB*, *NR1D1*, *UCP3*, *CD36*, *CASQ1*, *ACADM*, *SIK1*, *PDK2*, *RCAN1*, *PPP3CC*, and *ATP1B1*.

GO analysis of the blue2 module, which reflected the unique response of MS pigs to cold stress, revealed significant enrichment in various biological processes, including the cellular metabolic process, RNA processing, the primary metabolic process, and the nucleic acid metabolic process (Figure 6D,E). Within this module, *ACTC1*, *PRSS8*, *GPCPD1*, *NR4A2*, *SIRT6*, *PSMD14*, *PSMC6*, *NCBP2*, and *PYM1* were enriched in the metabolic process (Appendix A), while *IL18* was significantly enriched in the apoptotic signaling pathway. Regarding the KEGG analysis, the blue2 module was mainly enriched in pathways related to heat production, energy metabolism, and viral infection, including oxidative phosphorylation, thermogenesis, non-alcoholic fatty liver disease, metabolic pathways, and Epstein-Barr virus infection (Figure 6F). Specifically, *SIRT6* was enriched in thermogenesis (Appendix A), while *PSMD14* and *PSMC6* were enriched in Epstein-Barr virus infection, and *NCBP2* and *PYM1* were enriched in nucleocytoplasmic transport. Based on the combined results of the GO and KEGG analyses from the WGCNA, the candidate target genes of the blue2 module were identified as *ACTC1*, *PRSS8*, *GPCPD1*, *NR4A2*, *SIRT6*, *PSMD14*, *PSMC6*, *NCBP2*, *PYM1*, and *IL18*.

The pink module exhibited only seven intersection genes with the cold-response gene set shared by both LW and MS pigs (Figure 6G). On the other hand, the blue2 module showed four intersection genes with the cold-response gene sets shared by LW and MS pigs, and six intersection genes with the cold-response gene set unique to MS pigs. Altogether, there were 17 intersection genes in the pink and blue2 modules, which overlapped with the cold-responsive gene sets shared by LW and MS pigs, as well as the gene set specific to MS pigs. Further analysis established that the 17 intersection genes were Venn with the candidate genes obtained from the GO and KEGG analyses based on the pink module and the cold-response gene set shared by LW and MS pigs, respectively. The results indicated that *ARRDC3*, *ADRB2*, *CEBPB*, *NR1D1*, and *SIK1* were the cold-response genes shared by both LW and MS pigs, as identified by both methods, and these candidate genes were significantly enriched in pathways related to energy metabolism and heat regulation in the pink module. Similarly, the 17 intersection genes were Venn with the candidate genes obtained by the GO and KEGG analyses based on the blue2 module and the cold-response gene set specific to MS pigs, respectively. The results showed that *ACTC1*, *PRSS8*, and *IL18* were the specific cold-response genes of MS pigs, as determined by both methods, and these candidate genes were significantly enriched in pathways related to energy metabolism, heat production, and immune regulation in the blue2 module. Therefore, we ultimately selected *ARRDC3*, *ADRB2*, *CEBPB*, *NR1D1*, and *SIK1* as the cold-response candidate genes shared by LW and MS pigs, and *ACTC1, PRSS8*, and *IL18* as the specific cold-response candidate genes in MS pigs.

### 2.8. Validation Results of Common and Specific Cold Response Candidate Genes in Pig Longissimus Dorsi Muscle and Satellite Cells

The relative expression change trend in the eight candidate genes in qRT-PCR was basically consistent with the change trend observed in the FPKM of RNA sequencing, indicating the reliability of the sequencing results (Figure 7A). However, the consistent correlation coefficient (CCC) results showed that the consistency between the FPKM value and qRT-PCR value was poor, using the MedCalc software version 18 (MedCalc Software, Ostend, Belgium). Additionally, the qRT-PCR results of the thermogenesis and energy metabolism in the longissimus dorsi muscle of LW and MS pigs under cold stress demonstrated a significant increase (*p* < 0.05) in the *SERCAα2*, *SLN*, *CASQ2*, *UCP3*, and *PGC*-*1α* heat-related genes in both breeds (Appendix A). Cold stress was also found to significantly promote (*p* < 0.05) the expression of energy metabolism genes, including *PFK*, *PKM, LDHA*, *LDHA*, *HADHB*, and *CPT1B* in LW and MS pigs. The WB results showed that CASQ2, HADHB, PFK1, and other heat-related proteins were significantly increased (*p* < 0.05) except UCP3 in LW and MS pigs under cold stress (Figure 7B and Appendix A). Of note, the UCP3 expression in MS pigs was most significantly (*p* < 0.01) higher than that in LW pigs at both 25 °C and 4 °C, suggesting a higher mitochondrial oxidative phosphorylation level in MS pigs.

The qRT-PCR results of eight candidate genes in porcine skeletal muscle satellite cells under cold stress revealed a significant up-regulation (*p* < 0.05) in the 32 °C cold exposure group compared with the 37 °C normal temperature group; this finding was consistent with the changing trend in the genes in RNA-seq (Figure 7C). For cell function exploration, we selected the *PRSS8* gene, which specifically responds to cold stress in MS pigs. The interference efficiency of qRT-PCR revealed that the sus-si-PRSS8-1, sus-si-PRSS8-2, and sus-si-PRSS8-3 groups significantly decreased (*p* < 0.05) the expression of *PRSS8* (Figure 7D), making them suitable for further experiments. The qRT-PCR results demonstrated that interfering with *PRSS8* in porcine skeletal muscle satellite cells significantly down-regulated (*p* < 0.05) the expression of heat production genes, including *SERCAα2*, *SLN*, *UCP3*, and *PGC*-*1α* (Figure 7E). Moreover, the interference with *PRSS8* also led to a significant decrease (*p* < 0.05) in the expression of energy metabolism genes, such as *GLUT1*, *GLUT4*, *PFK*, *PKM*, *HADHB*, and *CPT1B*. In addition, WB analysis highlighted that interfering with *PRSS8* in porcine skeletal muscle satellite cells significantly reduced (*p* < 0.05) the expression of SERCAα2, CASQ2, UCP3, and PGC-1α heat production proteins (Figure 7F and Appendix A). The interference with *PRSS8* also led to a significant decrease (*p* < 0.05) in CPT1B protein levels, indicating that fatty acid oxidation was inhibited. Moreover, the level of extracellular signal-regulated kinase (ERK) phosphorylation was significantly decreased (*p* < 0.05) following the interference of *PRSS8*.

## 3. Discussion

Animals exhibit various adaptive responses under cold stress, including increased respiratory rate, decreased body temperature, diminished feed intake, and increased heat production in low-temperature environments. Pigs convert most of the nutrients in their diet into heat energy to maintain their body temperature and compensate for heat loss caused by the cold; this subsequently results in a decreased body weight and feed conversion rate [31,32]. Previous studies have shown that the cold tolerance of Chinese Meishan pigs and European LW pigs is closely associated with body weight and increases significantly with age (*p* < 0.01). However, Meishan piglets have 16% lower body weight than LW piglets, yet both exhibit a similar cold tolerance [33]. The findings of this study also support the notion that cold stress leads to weight loss in both LW and MS pigs, though it may not be significant due to the short duration of the cold stress. Additionally, the body weight of MS pigs was lower than that of LW pigs, yet they exhibited better cold adaptability. A study involving piglets (~13 days old) subjected to five days of cold stress observed a decrease in their core temperature and an increase in their cold adaptability, achieved through postural adjustments [34]. Another study on piglets after birth found that their tremor intensity gradually decreased under cold stress, while their heat production and muscle blood flow gradually increased [35,36]. The initial explanation for the decrease in trembling intensity attributed it to an increase in trembling efficiency [37]. However, current research suggests that the decrease is due to increased NST in muscles [23]. The results of the present study also revealed that cold stress significantly decreased (*p* < 0.05) the body temperature of both LW and MS pigs, in addition to significantly increasing (*p* < 0.05) the shivering frequency. However, on the fourth day of cold stress, the shivering frequency of MS pigs decreased significantly (*p* < 0.05), and their body temperature deviation was small, implying that MS pigs exhibited superior cold adaptability.

Cold stress triggers a series of adaptive responses in animals, involving the regulation of brain hormones sent to skeletal muscle for specific cold adaptations [38]. NE, the main neurotransmitter, is secreted under cold stress to mediate skin vasoconstriction and reduce heat loss [39]. Cold exposure influences systemic energy metabolism, activating BAT to mediate NST and reduce the plasma triglyceride concentration [40]. In piglets, cold stress was found to significantly increase the activities of glycolytic potential (GP) (*p* < 0.001) and LDH (*p* = 0.03), along with the activities of CS and β-hydroxy-acyl-CoA dehydrogenase (HAD), indicating increased glycolysis and oxidative metabolism [41]. The current study also examined serum hormones and enzyme activities, evidencing that cold stress significantly increased (*p* < 0.05) the activities of EPI, NE, Ca^2+^/Mg^2+^-ATPase, LDH, PFK-1, HK2, M2-PK, SDH, and CS in LW pigs. Conversely, there were no significant changes in the thermogenic and glucose metabolism enzymes of MS pigs under cold stress, except for a marked increase (*p* < 0.01) in the SDH and CS mitochondrial energy metabolism enzymes. Notably, at 4 °C, the amounts of SDH and CS mitochondrial energy metabolism enzymes were statistically significantly higher (*p* < 0.01) than those of LW pigs. Enzyme activity staining of the longissimus dorsi muscle in LW and MS pigs also demonstrated that the LW pigs responded to cold stress by enhancing their mitochondrial energy metabolism and glycolysis, while the MS pigs primarily increased their mitochondrial activity. Electron microscopy of the longissimus dorsi muscle revealed an increase in mitochondrial activity in response to cold stress by elevating the mitochondrial abundance in skeletal muscle. Of note, the skeletal muscle of the MS pigs exhibited a significantly higher mitochondrial abundance (*p* < 0.05) compared to the LW pigs at both 25 °C and 4 °C; furthermore, the mitochondrial ridge structure of the MS pigs under cold stress appeared more intact, indicating their superior ability to carry out mitochondrial energy metabolism.

Analyzing the entire mRNA transcriptome of cells, tissues, or organisms has proven to be a valuable approach for identifying specific genes in the response of animals to cold stress. Further, transcriptome sequencing is an increasingly useful tool for uncovering key pathways and biological processes associated with how animals adapt to changes in body temperature [38]. Earlier research has highlighted the increased transcriptional plasticity observed in deer mice [42] and chickens [43] in response to cold stress. In the current study, it was found that the number of DEGs and enrichment pathways in the MS pigs under cold stress were notably lower compared to the LW pigs, suggesting that the MS pigs were less affected by cold stress and exhibited a certain level of cold adaptability. The transcriptome analysis of Songpu mirror carp and bar carp adapted to 28 °C and 18 °C revealed a distinct cold tolerance between the larvae of the two breeds. The study identified several biological processes, including morphogenesis, secondary alcohol metabolism, drug transport, and the FoxO signaling pathway, which were closely associated with the development of cold tolerance and adaptability in these different breeds [44]. The transcriptional characteristics of inguinal fat in mice under cold stress showed that the inguinal white fat was primarily engaged in fatty acid elongation, as well as three acylglycerols’, sheath fat, and triglyceride synthesis pathways following a brief cold exposure at 4 °C for three days [45]. The transcriptome sequencing analysis of modern tropical and subtropical elephants highlighted the presence of DEGs that generated single nucleotide polymorphisms, fragment insertions, deletions, and amplifications of intact gene regions. These genes were found to be involved in lipid metabolism and thermogenesis, aiding elephants in coping with harsh and cold environments [46]. In a transcriptome study of grizzly bears during hibernation, the expression of fatty acid synthase and acetyl-CoA carboxylase, which were rate-limiting enzymes in long-chain fatty acid synthesis, decreased in the fat and muscle tissues of bears [47]. However, there was an increased expression of lipoprotein lipase, which acts as a negative regulator of hormone-sensitive lipase, triglyceride lipase, and monosaccharide lipase. This increase indicated that lipolysis was indirectly activated during hibernation. In the present study, the DEGs of LW and MS pigs in response to cold stress were found to be involved in various biological processes, including cold-induced thermogenesis, the lipid metabolic process, the carbohydrate biosynthetic process, the AMPK signaling pathway, and non-alcoholic fatty liver disease. These findings were consistent with previous research, highlighting that cold exposure significantly enhances thermogenesis and glycolipid energy metabolism in both pig breeds. Moreover, beyond regulating heat production and energy metabolism, MS pigs exhibited additional adaptations under cold stress. These adaptations involved the regulation of the hemoglobin metabolic process, oxygen transport, their adaptive immune response, transport, and their responses to influenza A.

The WGCNA method was employed to identify 13 gene co-expression modules in bees in response to cold stress (20 °C), representing different developmental stages and stress responses [48]. Among these modules, the dark orange module was found to be associated with autophagy-animal, endocytosis, and mitogen-activated protein kinase (MAPK) signaling pathways, suggesting that these pathways may play a crucial role in helping bees withstand low temperatures. Similarly, in plateau and plain yaks, four functional modules were identified in the DEGs through WGCNA analysis [49]. The turquoise and brown modules presented a high correlation with phenotypic traits such as weight, cooked meat percentage, and meat color. Furthermore, the transcriptome analysis of Chinese white wax scale insects under cold stress revealed their involvement in alcohol formation activity, lipid metabolism, membrane and structure maintenance, and oxidoreductase activity. The WGCNA results of Chinese white wax scale insects further determined that the genes within the module were associated with cytoskeletal proteins, the cytoskeletal protein pathway, the biosynthesis of unsaturated fatty acids, and glycerophospholipid metabolism [50]. Moreover, the tan module was found to contain hub genes, with the most prominent ones being *heat shock protein 10* (*hsp10*), *hsp60*, *hsp70*, and *hsp90*. In this study, the WGCNA analysis revealed the pink module to be enriched in several key biological processes, including adaptive thermogenesis, gluconeogenesis regulation, calcium-mediated signaling, oxidative phosphorylation, carbon metabolism, non-alcoholic fatty liver disease, the AMPK signaling pathway, and pyruvate metabolism. These findings connoted that both the LW and MS pigs responded to cold stress mainly by enhancing their heat production and glycolipid energy metabolism, which aligned with previous research findings. The blue2 module exhibited enrichment in metabolic processes, the apoptotic signaling pathway, oxidative phosphorylation, thermogenesis, Epstein-Barr virus infection, fatty acid oxidation, and immune pathways. These outcomes suggest that the specific response of MS pigs to cold stress is not limited to heat production and energy metabolism pathways but also involves the modulation of inflammation and immune-related pathways. This implies that MS pigs may possess enhanced cold adaptability through the regulation of their immune responses.

Phosphoproteins and SLN regulate SERCA pump activity, and their interaction with SERCA alters the kinetics of the calcium cycle, promoting thermogenesis [51,52]. *UCP3*, on the other hand, has been identified as a mediator of NST in pigs, as it dissipates proton gradients in mitochondria [9]. Recent murine studies have shown that cold exposure increases systemic energy expenditure and improves glucose metabolism by inducing sympathetic nervous system activity and recruiting brown adipocytes [53]. Additionally, cold exposure has demonstrated numerous effects on liver lipid metabolism and microbiome composition, contributing to the maintenance of systemic metabolic homeostasis along with BAT in mice [54]. In line with these findings, our study demonstrated that cold stress promotes inefficient calcium cycle thermogenesis, UCP3 uncoupling thermogenesis, and glycolipid energy metabolism in LW and MS pigs at the mRNA and protein levels. Moreover, the MS pigs displayed a relatively higher fatty acid oxidation ability.

In this study, five candidate genes (*ARRDC3*, *ADRB2*, *CEBPB*, *NR1D1*, and *SIK1*) were identified in response to cold stress in LW and MS pigs. These candidate genes were initially selected based on an enrichment analysis of the pink module, and LW and MS pigs shared a cold-response differential gene set which showed significant enrichment in dominant GO terms and KEGG pathways, mainly involving processes such as cellular metabolism, adaptive thermogenesis, viral infection, oxidative phosphorylation, the AMPK signaling pathway, and other processes. *Arrestin domain*-*containing protein 3* (*ARRDC3*) is known to contribute to the regulation of the mouse β-adrenergic receptor signaling pathway, alongside influencing the energy consumption of brown and white adipose tissue [55,56]. *Adrenoceptor beta 2* (*ADRB2*) specifically binds to endogenous ligands, including EPI and NE, to regulate intracellular cAMP levels. Furthermore, *ADRB2* activation catalyzes the exchange of GDP and GTP on G protein-coupled receptors and activates adenylate cyclase, converting ATP into cAMP, thereby activating the Raf/MAPK/ERK signaling pathway [57]. *CCAAT enhancer binding protein beta* (*CEBPB*) is integral in the adipogenic differentiation process and facilitates the regulation of PPAR expression [58,59]. Studies have concluded that acute cold exposure stimulates adrenaline release and leads to an increase in *CEBPB* mRNA levels in the brown fat of rats. Treating brown adipocytes with NE also results in an elevation of *CEBPB* mRNA levels [60]. *Nuclear receptor subfamily 1 group D member 1* (*NR1D1*) is a critical component of the mammalian circadian clock system. Its endogenous ligand is heme, a metabolite involved in mitochondrial respiration and cellular redox balance, making *NR1D1* serve as a sensor for cellular metabolism [61]. *Salt*-*inducible kinases* (*SIKs*) are members of the AMPK family and play an important regulatory role in glycolipid metabolism [62,63]. Previous research has shown that manipulating *SIK1* expression can impact gluconeogenesis [64] and fat metabolism in the liver of mice [65]. Silencing *SIK1* leads to the up-regulation of genes involved in adipogenesis, whereas overexpression has the opposite effect, suggesting a specific regulatory mechanism for *SIK1* in modulating metabolic processes related to adipocyte formation and fat metabolism. The current study observed that five candidate genes shared by LW and MS pigs were up-regulated in response to cold stress in both pig longissimus dorsi and porcine skeletal muscle satellite cells. These findings imply that both pig breeds employ a similar response mechanism to cope with cold stress, involving an increase in heat production and glycolipid energy metabolism.

*ACTC1*, *PRSS8*, and *IL18* were identified as candidate genes specifically responding to cold stress in MS pigs. These candidate genes were initially selected based on enrichment analysis of both the blue2 module and the MS pig-specific cold-response differential gene set, which both showed significant enrichment in dominant GO terms and KEGG pathways, mainly involving processes such as metabolic process, heat production, energy metabolism, viral infection, and other processes. *Actin alpha cardiac muscle 1* (*ACTC1*), a cardiac α-actin, serves as the main protein of cardiac myofilament, responsible for the heart’s systolic function [66]. Studies in mice have demonstrated that the overexpression of *ACTC1* can alleviate muscle dysplasia caused by skeletal muscle α-actin deficiency [67]. Interestingly, the amount of *ACTC1* and *MHCB* transcripts in chickens remained unchanged under cold stimulation [68], whereas the regulation of the *ACTC1* gene in mammals appears to be strongly affected by cold stress [69]. In Altay lambs, known for their cold adaptability, the expression of *PVALB*, *TNNC1*, *MYL2*, and *ACTC1* genes related to muscle contraction was observed to be higher compared to Hu lambs under cold stress [70]. *Interleukin*-*18* (*IL*-*18*), initially believed to stimulate Th1 cells to produce IFN-γ through CD3, especially in the presence of *IL*-*12*, has been found to have additional functions [71]. IL-18, in conjunction with IL-3, induces mast cells and basophils to produce IL-4 and IL-13, thereby stimulating both innate and acquired immunity [72]. Moreover, IL-18 induces the phosphorylation of PI3K/Akt/S6 and mammalian target of rapamycin (mTOR), which in turn influences the expression of *Bcl*-*xL* and *Bcl2* [73,74]. *Protease serine 8* (*PRSS8*), also known as prostasin, is a serine protease that is a vital element of various biological processes, including protein degradation and digestion, protein processing, and tissue remodeling. Recent research indicates that *PRSS8* can influence liver sensitivity to insulin through its involvement in regulating TLR4-mediated signaling pathways [75]. When *PRSS8* is overexpressed in the liver, it leads to the phosphorylation of the ERK pathway, resulting in improved glucose and lipid metabolism, which can help ameliorate fatty liver disorders [76]. In this study, three candidate genes that are specifically responsive to cold stress in MS pigs showed increased expression in pig longissimus dorsi and skeletal muscle satellite cells under cold stress. This indicated that MS pigs, when exposed to cold stress, not only enhance their heat production and glycolipid energy metabolism but that they also rely on enhancing their immune response to bolster their body resistance and achieve better cold adaptability. This study also found that the knockdown of *PRSS8* in porcine skeletal muscle satellite cells had a significant down-regulatory effect on genes and proteins related to thermogenesis, glucose metabolism, and fatty acid oxidation. Furthermore, it inhibited ERK phosphorylation and reduced energy metabolism. These results are corroborated by functional studies of *PRSS8* in other animals, demonstrating a positive correlation between *PRSS8* and thermogenesis as well as energy metabolism.

## 4. Materials and Methods

### 4.1. Animals

The current study was conducted in the Department of Animal Science, Shanxi Agricultural University. The research protocol for the present experiment was approved by the Care and Use Committee of Shanxi Agricultural University (license number: SXAU-EAW-2021MS.P.052801). The method was based on the “Guidelines for the Care and Use of Laboratory Animals” of the Ministry of Agriculture.

A total of 12 castrated male MS pigs and LW pigs, aged 90 days, were selected from the Datong pig breeding farm (Datong, China). Pigs that are 90 days old are in a phase of rapid growth and development, which serves as a crucial period in the pig’s overall growth process. At the same time, this stage is in the middle period of the early stage of growth and development at 60 days of age and the late stage of growth and development at 120 days of age, and also belongs to the middle period of birth at 180 days of age, so this stage is a key period in the growth and development of pigs. Therefore, we can better understand the physiological and transcriptional changes in pigs in a low temperature stress environment by selecting pigs at the age of 90 days for cold stress experiments. They were divided into four groups: MS pig normal-temperature group (MS-25), MS pig low-temperature group (MS-4), LW pig normal-temperature group (LW-25), and LW pig low-temperature group (LW-4), with three pigs in each group. The pigs were reared under normal temperature (25 ± 1 °C) and low temperature (4 ± 1 °C), respectively. For the experiment, the pigs were kept in an environment control chamber, with each artificial climate chamber operating on a 12 h day and night cycle. A preparatory feeding period of 7 days was followed by a 4-day experimental period. Throughout the experiment, the pigs had ad libitum access to feed and water.

### 4.2. Phenotypic Detection and Sample Collection

During the experiment, each pig underwent three daily measurements at 3 p.m., including recording the core temperature, calculating the respiratory rate based on abdominal fluctuations per minute, and determining the shivering frequency by counting shivers per minute. Blood samples were collected from the anterior vena cava of each pig at both the beginning and end of the experiment using anticoagulant-free tubes. After the experiment concluded, the animals were anesthetized using electric shock, and the carotid artery was immediately severed to ensure euthanasia. Aseptic techniques were employed to obtain the longissimus dorsi muscle from the pig. The collected muscle was promptly subdivided into cryopreservation tubes and immediately placed in liquid nitrogen for rapid freezing. The tubes were then stored at −80 °C in a refrigerator. Additionally, the 1 cm^3^ sections of the longissimus dorsi muscle were wrapped in tin foil paper coated with optimal cutting temperature compound and immediately placed in liquid nitrogen. Furthermore, 1 mm^3^ segments of the longissimus dorsi muscle were excised and placed in Eppendorf (EP) tubes containing 2.5% glutaraldehyde. The tubes were then stored at 4 °C in a refrigerator for further processing.

### 4.3. Serum Biochemistry and Hormone Analyses

Enzyme activities related to various metabolic processes were assessed using ELISA kits (Mlbio, Shanghai, China). Specific detection kits were employed to measure the levels of heat-related hormones and enzyme activities, including norepinephrine (NE), epinephrine (EPI), and Ca^2+^/Mg^2+^-ATPase. In addition, enzyme activities associated with glucose metabolism, including lactate dehydrogenase (LDH), phosphofructokinase 1 (PFK1), hexokinase 2 (HK2), and pyruvate kinase M2 isoenzyme (M2-PK), were analyzed using the respective detection kit. Furthermore, mitochondrial energy metabolism-related enzyme activity was examined using a citrate synthase (CS) and succinate dehydrogenase (SDH) detection kit.

### 4.4. Electron Microscopy and Immunohistochemistry

The longissimus dorsi muscle tissues were stored in EP tubes containing 2.5% glutaraldehyde and fixed in a 1% OsO_4_ solution for one hour. Subsequently, the samples were processed through infiltration, embedding, slicing, and staining with 2% uranyl acetate and lead citrate to obtain ultrathin sections. The images were then observed and captured by transmission electron microscopy (HT7800, Hitachi, Tokyo, Japan). This allowed for the visualization of mitochondrial abundance and structural integrity in the longissimus dorsi muscle.

The freezing microtome was pre-cooled to −20 °C beforehand, and the samples were allowed to equilibrate for 20 min to reach a stable temperature. Tissue sections were obtained with a fine adjustment of 7 μm to achieve the desired thickness. These sections were subsequently carefully placed onto prepared slides. Following the evaporation of tissue fog, the slides were examined under a microscope to assess their morphology. Slides displaying intact tissue structure were carefully chosen and stored at −80 °C in a refrigerator. The sections of porcine longissimus dorsi were subjected to histochemical staining using commercially available SDH (Solarbio, Beijing, China) and LDH (Solarbio, Beijing, China) staining kits, following the provided instructions. The histochemical staining intensity of SDH and LDH served as indicators of the tissue’s ability to regulate mitochondrial energy metabolism and glycolysis, respectively. Image analyses were performed using the Image-Pro Plus system (Version 6.0, Media Cybernetics, Silver Spring, MD, USA).

### 4.5. RNA-Seq

Transcriptome sequencing of the longissimus dorsi muscle from 12 pigs was conducted by Gene Denovo (Gene Denovo Co., Ltd., Guangzhou, China). The RNA samples with high purity (OD260/280 ≥ 2.0) and high integrity (RIN > 8) were sequenced using Illumina Novaseq 6000, resulting in a substantial number of original reads. Fastp was employed to ensure the quality of the raw data, and high-quality clean reads were obtained by filtering out reads containing more than 10% unknown nucleotides (N) and more than 50% low-quality bases (Q value ≤ 20). StringTie software (v2.2.0) was used to calculate fragments per kilobase of transcript per million mapped reads (FPKM) to quantify the expression abundance of and variation in each transcript. DESeq2 software (version 3.0) was utilized for analysis to identify DEGs between the two groups. The screening threshold was set as log_2_|FoldChange| ≥ 1, with *p* < 0.05. To provide functional insights, the DEGs were annotated to GO terms and KEGG pathways using annotation and visualization methods. The abundance of DEGs was normalized based on the z-score and represented by −log_10_*p* value and fold enrichment. The sequencing data were uploaded to NCBI (accession: PRJNA999915).

### 4.6. Weighted Gene Co-Expression Network Analysis

For the analysis of all gene expression data, the WGCNA R package (v1.70) was used to construct a weighted gene co-expression network. The dynamic segmentation method was applied to identify and classify modules according to their expression patterns. The following parameters were set: cut height = 0.8, minsize = 10. Default values were used for other parameters. Eigenvalue analysis was carried out to assess the correlation between modules and sample groups. Further, GO and KEGG enrichment analyses were performed based on the genes within the target modules.

### 4.7. Cell Culture

Porcine skeletal muscle satellite cells were retrieved from a liquid nitrogen tank and rapidly thawed in a 37 °C water bath. After thawing, the cells were immediately centrifuged at 800 r/min at room temperature for five minutes using a centrifuge (5810R, Eppendorf, Hamburg, Germany). The supernatant was then discarded, and the cells were resuspended in a complete medium consisting of 10% fetal bovine serum, 1% penicillin-streptomycin, and high glucose Dulbecco’s Modified Eagle Medium (DMEM) before being inoculated into 6-well plates. Using the porcine protease serine 8 (PRSS8) gene sequence as a reference, four siRNA sequences were custom-synthesized by RiboBio, (RiboBio Co., Ltd., Guangzhou, China), and the sequence information is provided in the Appendix A. The transfection of the four siRNAs into porcine skeletal muscle satellite cells was accomplished using LipofectamineTM 2000 (Thermo Fisher Scientific, Waltham, MA, US) when cell fusion reached approximately 70%. Following transfection, the complete medium was replaced and maintained for 6–12 h. Upon achieving around 90% cell fusion, the differentiation medium (2% HS, 1% penicillin-streptomycin, high glucose DMEM) was introduced to induce myogenic differentiation, and the medium was refreshed every 2 days. After 6 days of differentiation, cells were harvested for the evaluation of heat production and energy metabolism marker gene expressions. For the cold exposure experiment, both untransfected and differentiated porcine skeletal muscle satellite cells were assigned to two groups: a control group maintained at 37 ± 0.5 °C and a cold exposure group at 32 ± 0.5 °C for 8 h. It was previously determined that 32 °C represents clinically mild hypothermia levels [77,78].

### 4.8. RNA Preparation and Quantitative Real-Time Polymerase Chain Reaction

Total RNA was extracted from skeletal muscle tissue using RNAiso (Takara, Tokyo, Japan) according to the manufacturer’s instructions. The quality and purity of the total RNA were evaluated at 260 and 280 nm using a spectrophotometer (NanoDrop 2000, Thermo Fisher Scientific, Waltham, MA, USA). Meanwhile, the absorbance ratio (260/280 nm) of the RNA extracted from all the samples was between 1.80 and 2.00. Subsequently, 1–2 μg of total RNA was subjected to reverse transcription, generating complementary DNA (cDNA) with the QuantiTect Reverse Transcription Kit (Qiagen GmbH, Hilden, Germany). The cDNA was stored at −20 °C in a refrigerator. For qRT-PCR, SYBR Premix Ex Taq (Takara, Kyoto, Japan) was used with a 10× dilution of the cDNA. Each sample was subjected to three technical replicates for accurate analysis. We used the no-template negative controls, and enzyme-free water was used as a substitute for nucleic acids in the qRT-PCR reaction. Other reagents were added in their normal proportions to monitor any contamination in the reaction system. The qRT-PCR protocol was as follows: 95 °C for 5 min, followed by 40 cycles of 95 °C for 5 s and 60 °C for 45 s. The primers for each gene were designed based on the available mRNA sequences in GenBank, utilizing the Primer-BLAST module of the NCBI website. For normalization, *18S* was used as the housekeeping gene, and the primer information is provided in the Appendix A. These primers were synthesized by Bioengineering, (Bioengineering Co., Ltd., Shanghai, China). The relative mRNA expression levels were determined using the 2^−ΔΔCT^ calculation method.

### 4.9. Western Blotting

All skeletal muscle tissues were first ground into a fine powder in liquid nitrogen. The tissue powder and satellite cells were then lysed in total protein lysis buffer, containing radioimmunoprecipitation assay and phenylmethylsulphonyl fluoride, at a ratio of 100:1. The lysates were incubated on ice for 30 min. After that, the samples were centrifuged at 12,000 r/min at 4 °C for 15 min, and the resulting supernatants were collected. They were then denatured at 100 °C for 10 min with 5× sodium dodecyl sulfate (SDS) loading buffer at a ratio of 4:1. Equal amounts of protein were separated using a 10% sodium dodecyl sulfate-polyacrylamide gel electrophoresis (SDS-PAGE) gel, running at 80 V for 30 min and 120 V for 90 min. The proteins were transferred onto 0.45 μm polyvinylidene difluoride membranes (Millipore, Billerica, MA, USA) at 120 V for 90 min. Subsequently, the membranes were incubated in 5% skim milk powder solution at room temperature and gently shaken for 1 h to block non-specific binding. Primary antibodies, including SERCA2 (Bioss, Beijing, China), CASQ2 (Proteintech, Wuhan, China), UCP3 (Abcam, Cambridge, UK), PGC-1α (Proteintech, Wuhan, China), CPT1B (Proteintech, Wuhan, China), PRSS8 (Affinity Biosciences, Changzhou, China), ERK1/2 (Proteintech, Wuhan, China), Phospho-ERK1/2 (Proteintech, Wuhan, China), PFK1 (Bioss, Beijing, China), HADHB (Bioss, Beijing, China), β-actin (1:5000, Abcam, Cambridge, UK), and β-Tubulin (Affinity Biosciences, Changzhou, China), were used to probe the membranes either for 2 h at room temperature or overnight at 4 °C. After washing the membranes with 0.05% Tris-buffered saline containing Tween-20 buffer (TBST), appropriate infrared (IR)-linked secondary antibodies (LI-COR, Lincoln, NE, USA) were used to probe the membranes for 2 h at room temperature. Following another wash with 0.05% TBST, the membranes were scanned using the LI-COR Odyssey scanner, and the gray value of the protein bands was calculated and analyzed using Image J software Version 1.8 (National Institutes of Health, Bethesda, MD, USA).

### 4.10. Statistical Analysis

Statistical analysis was performed using IBM SPSS Statistics Version 26.0 (IBM Corp, Armonk, NY, USA) and GraphPad Prism Version 8.0 (GraphPad Software Inc., San Diego, CA, USA). To determine the significance of differences between two groups, the two-tailed unpaired Student’s *t*-test was employed. For analyzing the significance of differences among multiple groups, one-way analysis of variance (ANOVA) with Duncan’s multiple range test was employed. Statistically significant levels were set as ** *p* < 0.01 and * *p* < 0.05. All data were presented as mean ± standard error of mean (SEM).

## 5. Conclusions

LW and MS pigs exhibited a decrease in body temperature and body weight under cold stress, along with increased shivering and respiratory frequencies. However, despite these changes, the MS pigs maintained stable physiological homeostasis, demonstrating their cold adaptability. The LW pigs responded to cold stress chiefly by regulating their heat production, glucose metabolism, and fatty acid oxidation. Meanwhile, the MS pigs exhibited a specific response to cold stress, showcasing their strong fatty acid oxidation ability, mitochondrial energy metabolism, and immune response, which contributed to their enhanced adaptability. This study identified *ARRDC3*, *ADRB2*, *CEBPB*, *NR1D1*, and *SIK1* as common candidate genes that are responsive to cold stress in LW and MS pigs. In contrast, *ACTC1*, *PRSS8*, and *IL18* were identified as specific candidate genes that uniquely responded to cold stress in the MS pigs. Furthermore, it was deduced that *PRSS8* plays a considerable role in regulating heat production, glucose metabolism, and fatty acid oxidation marker genes and proteins in porcine skeletal muscle satellite cells. Moreover, *PRSS8* was found to influence energy metabolism through its impact on ERK phosphorylation. By elucidating the specific genetic and metabolic pathways involved in the response of LW and MS pigs to cold stress, this study contributes valuable insights that can help in developing strategies for enhancing the cold resistance and adaptability of pigs, ultimately leading to improved livestock management and increased economic gains in the pig industry.

## Figures and Tables

**Figure 1 ijms-24-15534-f001:**
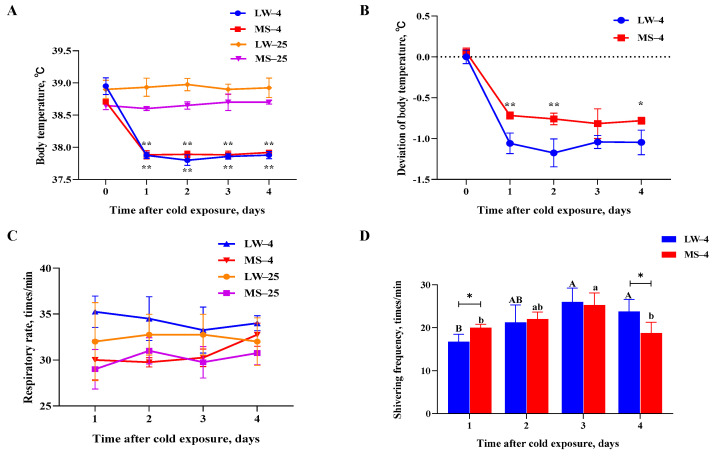
Physiological response patterns of (**A**) body temperature (significant difference analysis was performed on the same breed at different temperatures), (**B**) body temperature deviation, (**C**) respiratory rate, and (**D**) shivering frequency in LW and MS pigs exposed to temperatures of 25 °C and 4 °C for four days. The results were expressed as mean ± SEM (n = 3). Statistical significance was determined using the two-tailed unpaired Student’s *t*-test, with ** *p* < 0.01 and * *p* < 0.05 denoting significance. ^A,B^ Different capital letters represent significant differences in LW pigs at 4 °C (*p* < 0.05). ^a,b^ Different lowercase letters represent significant differences in MS pigs at 4 °C (*p* < 0.05). LW and MS represent Large White pigs and Mashen pigs, respectively; 25 and 4 indicate 25 °C and 4 °C, respectively.

**Figure 2 ijms-24-15534-f002:**
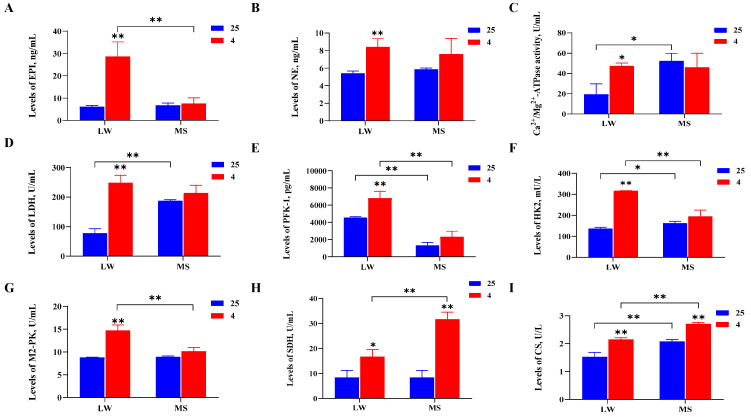
Serum biochemistry and hormone analyses of (**A**) epinephrine (EPI), (**B**) norepinephrine (NE), (**C**) Ca^2+^/Mg^2+^-ATPase, (**D**) lactate dehydrogenase (LDH), (**E**) phosphofructokinase 1 (PFK-1), (**F**) hexokinase 2 (HK2), (**G**) pyruvate kinase M2 isoenzyme (M2-PK), (**H**) succinate dehydrogenase (SDH), and (**I**) citrate synthase (CS) in LW and MS pigs at 25 °C and 4 °C. The results are presented as mean ± SEM (n = 3). LW and MS refer to Large White pigs and Mashen pigs, respectively; 25 and 4 refer to 25 °C and 4 °C, respectively. *p* values were determined using the two-tailed unpaired Student’s *t*-test, ** *p* < 0.01; * *p* < 0.05.

**Figure 3 ijms-24-15534-f003:**
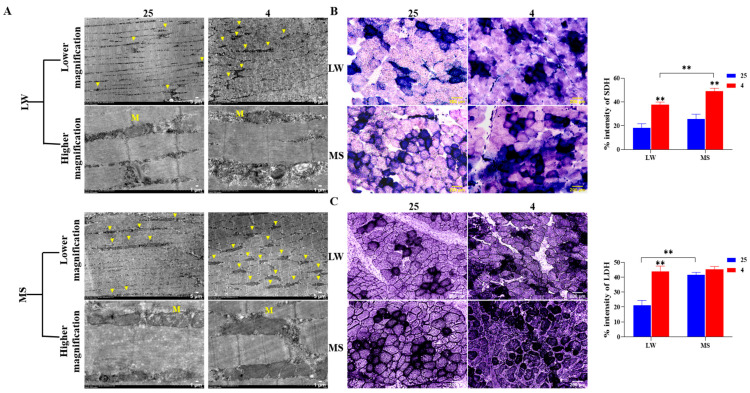
Effects of cold stress on the ultrastructure and enzyme activity of the longissimus dorsi muscle in LW and MS pigs at 25 °C and 4 °C. (**A**) Transmission electron micrographs of longissimus dorsi from LW and MS pigs at 25 °C and 4 °C, whereby the top panels and bottom panels show low (scale bar 5 μm) and high magnification (scale bar 1 μm), respectively. Yellow triangles and ‘M’ represent mitochondria. (**B**) Succinate dehydrogenase (SDH) activity staining and (**C**) lactate dehydrogenase (LDH) activity staining of longissimus dorsi muscle tissue in LW and MS pigs at 25 °C and 4 °C (scale bar 200 μm). The results are expressed as mean ± SEM (n = 3). LW and MS represent Large White pigs and Mashen pigs, respectively; 25 and 4 refer to 25 °C and 4 °C, respectively. *p* values were determined using the two-tailed unpaired Student’s *t*-test. ** *p* < 0.01.

**Figure 4 ijms-24-15534-f004:**
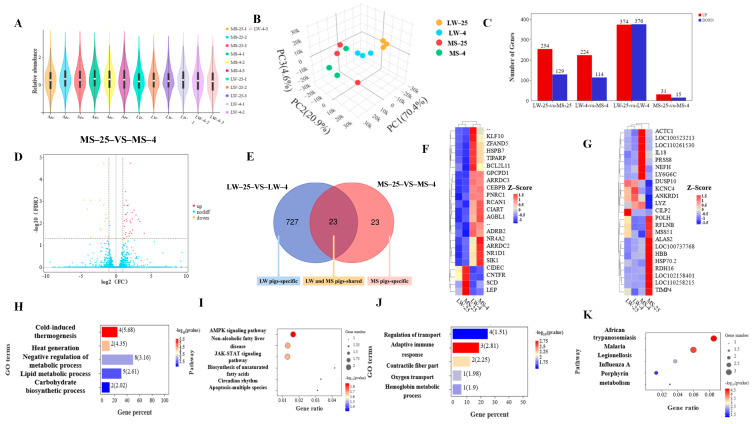
Transcriptomes of longissimus dorsi muscle in LW and MS pig at 25 °C and 4 °C. (**A**) Violin plot presenting the distribution of transcriptome sequencing samples from 12 pigs in LW and MS groups. The white dot on the violin plot represents the median, and the quartile range is shown by the black bar above and below the center of the violin. (**B**) The PCA plot of all 12 samples. (**C**) Histogram depicting the number of identifiable DEGs between the four groups. (**D**) Volcanic map highlighting the DEGs between MS-25 and MS-4 groups. (**E**) Venn diagram illustrating the DEGs shared between the LW pig group and the MS pig group at 25 °C and 4 °C. (**F**) Heatmap visualizing the DEGs shared in the LW pig group and the MS pig group at 25 °C and 4 °C. (**G**) Heatmap displaying the specific DEGs between MS-25 and MS-4 groups. (**H**) Gene ontology (GO) enrichment analysis of DEGs shared in the LW pig group and the MS pig group at 25 °C and 4 °C. (**I**) Kyoto encyclopedia of genes and genomes (KEGG) pathway classification of DEGs shared in the LW pig group and the MS pig group at 25 °C and 4 °C. (**J**) GO enrichment analysis of specific DEGs between MS-25 and MS-4 groups. (**K**) KEGG pathway classification of specific DEGs between MS-25 and MS-4 groups. LW and MS represent Large White pigs and Mashen pigs, respectively; 25 and 4 indicate 25 °C and 4 °C, respectively. The heat map illustrates the relative expression pattern of DEGs between groups, with each column representing a sample and each row representing the expression level of a single mRNA in different samples. The color range of the heat map ranges from blue (low expression) to red (high expression).

**Figure 5 ijms-24-15534-f005:**
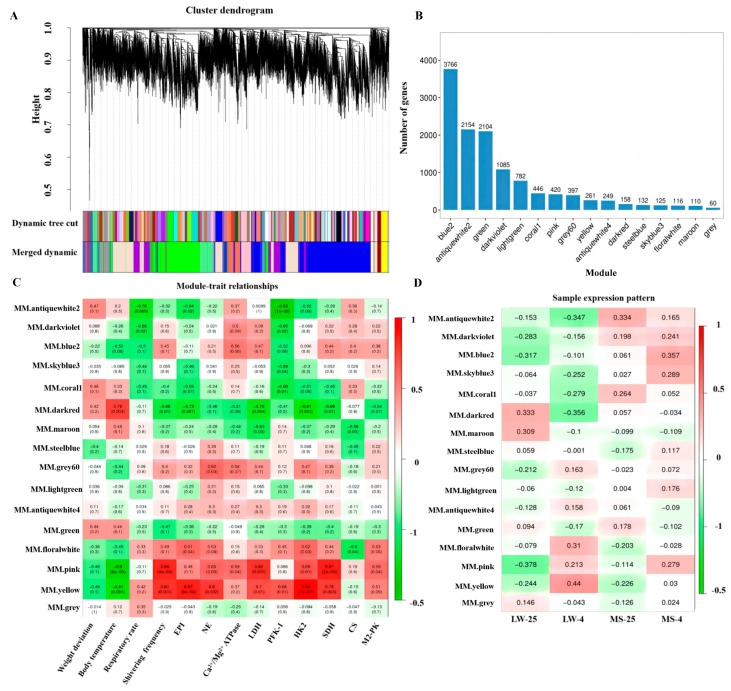
Weighted co-expression network analysis of all genes. (**A**) Gene hierarchical clustering partition module. Different colors represent different modules. (**B**) Histogram showing the number of genes in each module. (**C**) Correlation analysis between modules and phenotypes. (**D**) Correlation analysis between modules and groups. The color range of the heat map ranges from green (low expression) to red (high expression). LW and MS represent Large White pigs and Mashen pigs, respectively; 25 and 4 indicate 25 °C and 4 °C, respectively.

**Figure 6 ijms-24-15534-f006:**
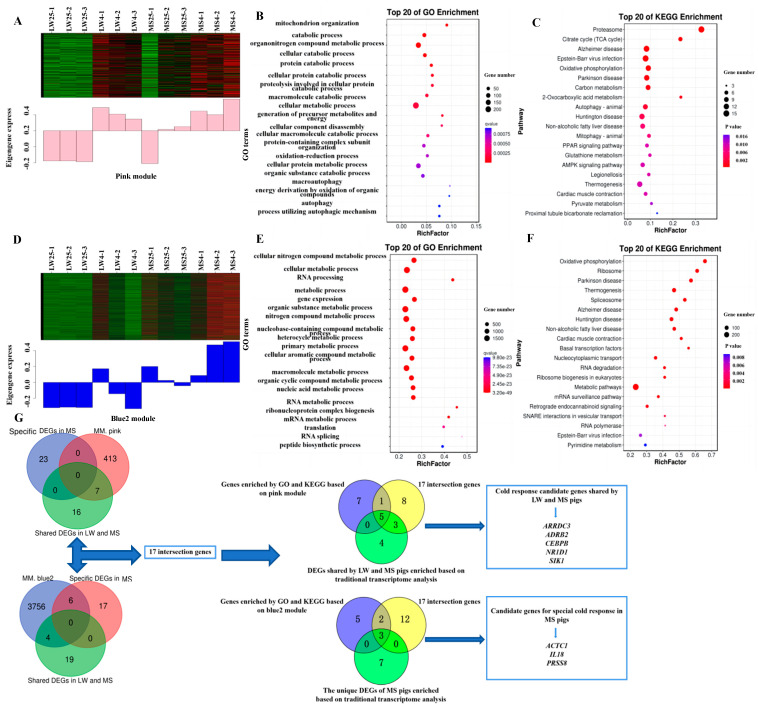
The expression patterns and functional analysis of genes in the co-expression modules. (**A**) The expression pattern of genes in the pink module. (**B**) Gene Ontology (GO) analysis of genes in the pink module. (**C**) Kyoto Encyclopedia of Genes and Genomes (KEGG) enrichment analysis of genes in the pink module. (**D**) The expression pattern of genes in the blue2 module. (**E**) GO analysis of genes in the blue2 module. (**F**) KEGG enrichment analysis of genes in the blue2 module. (**G**) Combined results of traditional transcriptomics analysis and weighted co-expression network analysis (WGCNA) to screen common and specific candidate genes responding to cold response in LW and MS pigs. The heat map color scale indicates expression levels, ranging from blue (low expression) to red (high expression). LW and MS denote Large White pigs and Mashen pigs, respectively; 25 and 4 represent 25 °C and 4 °C, respectively.

**Figure 7 ijms-24-15534-f007:**
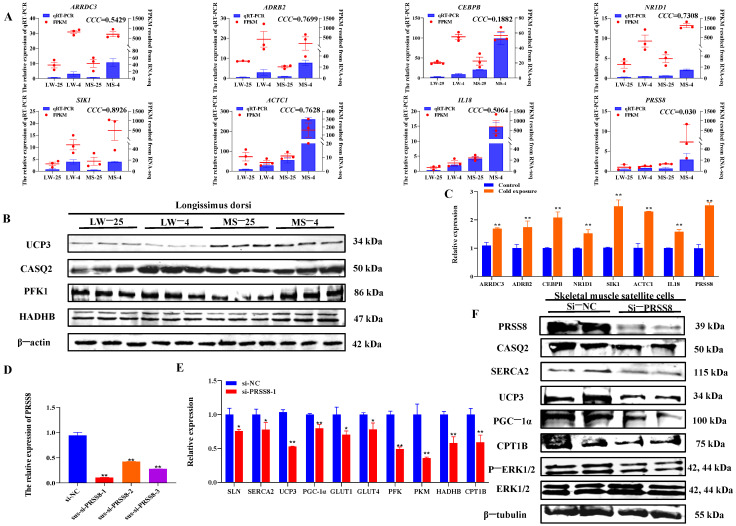
Validation results of common and specific cold-responsive candidate genes in pig longissimus dorsi muscle tissue and satellite cells. (**A**) qRT-PCR verification of candidate genes in the longissimus dorsi muscle tissue of LW and MS pigs at 25 °C and 4 °C. The left y-axis and blue bars represent the relative expression levels of eight candidate genes in qRT-PCR, and the right y-axis and red bars represent the FPKM values of eight candidate genes obtained from RNA sequencing. (**B**) Western blot verification of candidate genes in the longissimus dorsi muscle tissue of LW and MS pigs at 25 °C and 4 °C. (**C**) qRT-PCR verification of candidate genes in porcine skeletal muscle satellite cells at 37 °C and 32 °C. (**D**) Detection of interference efficiency of PRSS8 in porcine skeletal muscle satellite cells. (**E**) Effects of PRSS8 interference on mRNA levels of heat production and energy metabolism in porcine skeletal muscle satellite cells. (**F**) Effects of PRSS8 interference on protein levels of heat production and energy metabolism in porcine skeletal muscle satellite cells. *p* values were determined using the two-tailed unpaired Student’s *t*-test. ** *p* < 0.01; * *p* < 0.05. LW and MS denote Large White pigs and Mashen pigs, respectively; 25 and 4 represent 25 °C and 4 °C, respectively.

## Data Availability

All data generated or analyzed during this study are available from the corresponding authors upon reasonable request. The data are not publicly available due to the fact that we are conducting further experiments.

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
