# Peer review of "Transcriptome Analysis Revealed Potential Genes of Skeletal Muscle Thermogenesis in Mashen Pigs and Large White Pigs under Cold Stress"

_ijms, 2023, doi:10.3390/ijms242115534_

Round 1

Reviewer 1 Report

General comments

This manuscript aims to study the mechanisms of cold adaptation in two breeds, Large White and Mashen. The introduction adequately contextualizes the readers for the topic at hand and methods implemented are plenty but could be better described. The results and discussion sections are coherent and the conclusions appropriate. The manuscript is overall written in good and comprehensive English but most figures present in the manuscript lack quality and several of them are impossible to read. My detailed comments are listed below.

Specific comments:

Lines 73/74 – “Lasiopodomys brandtii should be italicized.

Line 118 – Please correct to “physiological”.

Lines 398/399 – It is mentioned that the expression trends of qPCR are “consistent” with the RNAseq results, how did the authors reach such conclusion if there is no statistical test involved? I recommend to the authors to apply the concordance correlation coefficient (CCC) which is normally used for these situations as seen in many related papers.

Line 658 – Did the authors not measure feed consumption? I think it would be an interesting variable to add to the study since low temperature reared pigs should be prone to eat more and that fact can influence a lot of variables.

Lines 749/750 – I think a reference should be added to support the claims that 32°C represent clinically mild hypothermia.

Lines 756/757 – The total RNA extraction protocol or kit is not mentioned anywhere.

Lines 762/763 – The authors do not mention the use of no template controls (negative controls) in the qPCR experiment. Furthermore, the authors do not mention the cycling conditions.

Author Response

We feel great thanks for your professional review work on our manuscript. As you are concerned, there are several problems that need to be addressed. According to your nice suggestions, we have made extensive corrections to our previous manuscript, the detailed corrections are given in the following point-by-point reply. In particular, we have improved and revised the quality and clarity of all figures in this manuscript. Furthermore, we have reviewed and restructured the article to enhance its logical flow and improve its readability. Thank you again for your positive comments and valuable suggestions to improve the quality of our manuscript.

Reviewer 2 Report

The authors investigated transcriptional changes related to cold stress in porcine skeletal muscles. The experimental design looks straightforward, and here are a couple of comments.

Line 48: Gene symbols and their full names should be in italics. Please address it throughout the manuscript.

Lines 98-112: This paragraph is about the method, so it needs to be moved or merged into the M&M section.

Line 123, “… most significantly or significantly …”: Please reword the sentence.

Line 133: Weight loss was greater in MS, which contradicts the interpretation that MS has greater adaptability to cold. The comprehensive interpretation across all measured traits is required in this section.

Line 185: The MDS plot or PCA plot using all 12 samples simultaneously would be more effective than the heat map using a specific DEG set to show group-specific similarities. I suggest adding such a figure.

Line 188, “~ 750 genes …”: Did ‘gene’ mean DEGs? Please make it clear.

Line 236, Figure 4 F and G: There is no legend for the colors. Is it a fold change?

Line 278: The figure numbers in Figure 4 need to be reordered. Fig. 4I was described after Fig 4J.

Line 334, the symbol of the correlation coefficient: R >> r in italic

Line 344: Why was not the yellow module selected?

Line 380: It looks like the number of DEGs overlapped with certain modules is relatively small compared to the number of genes of modules (n=413 and n=3756). It would be good to add some interpretation of how these genes are representative in the modules into the discussion section.

Line 397: The relevant reasons why these genes were selected as candidates should be described here.

Line 650: I would suggest adding the biological meaning of 90 days old for the experimental design for the cold stress.

Author Response

We appreciate the professional and valuable suggestions made of you, on the basis of which we have revised and strengthened our manuscript. In particular, we have added a detailed description of the biological significance of selecting 90-day-old pigs as subjects, and restructured the article to make it more logical and easy to understand. Changes are tracked and highlighted in the revised manuscript, and all our responses to the comments are detailed in the following point-by-point response. Thank you again for your positive comments and valuable suggestions to improve the quality of our manuscript.

Round 2

Reviewer 1 Report

Most of the issues have been addressed and the overall quality of the publication has improved. On the other hand, some of the pictures still lack quality and/or present graphical mistakes (such as the odd look of the celsius degree).

Author Response

For research article

Response to Reviewer 1Comments

Dear Reviewer:

Thank you for your comments regarding our manuscript entitled “Transcriptome analysis revealed potential genes of skeletal muscle thermogenesis in Mashen pigs and Large White pigs under cold stress” (Manuscript ID: ijms-2626786). We are sorry for the lack of quality and graphic errors of the pictures in this manuscript, which is indeed our deficiency. According to your nice suggestions, we have improved and revised the quality and clarity of all the pictures in this manuscript, especially the Celsius degree. We again carefully examined the manuscript and gave the changes/additions to the manuscript in the red text. We sincerely thank you again for your valuable feedback that we have used to improve the quality of our manuscript.

Reviewer 2 Report

The authors have addressed the initial concerns presented in my initial review.

Author Response

For research article

Response to Reviewer 2 Comments

Dear Reviewer:

Thank you for your comments regarding our manuscript entitled “Transcriptome analysis revealed potential genes of skeletal muscle thermogenesis in Mashen pigs and Large White pigs under cold stress” (Manuscript ID: ijms-2626786). Thank you very much for agreeing to sign this manuscript evaluation report. We again carefully examined the manuscript and gave the changes / additions to the manuscript in the red text. We feel great thanks again for your positive comments and valuable suggestions to improve the quality of our manuscript. 
